# The Knowledge Gap in Gut Microbiome Characterization in Early-Onset Colorectal Cancer Patients: A Systematic Scoping Review

**DOI:** 10.3390/cancers17111863

**Published:** 2025-05-31

**Authors:** Rita Gomes de Sousa, Catarina Sousa Guerreiro, Inês Santos, Marília Cravo

**Affiliations:** 1Gastroenterology Department, Hospital da Luz Lisboa, 1500-650 Lisboa, Portugal; marilia.cravo@sapo.pt; 2Nutrition Department, Hospital da Luz Lisboa, 1500-650 Lisboa, Portugal; cfguerreiro@medicina.ulisboa.pt; 3Faculdade de Medicina, Universidade de Lisboa, 1649-028 Lisboa, Portugal; santosi@medicina.ulisboa.pt; 4Instituto de Saúde Ambiental (ISAMB), Faculdade de Medicina, Universidade de Lisboa, 1649-028 Lisboa, Portugal

**Keywords:** colorectal cancer, human microbiome, scoping review, young adults

## Abstract

The incidence of early-onset colorectal cancer (EoCRC) is rising globally, yet the underlying causes of this increase remain poorly understood. Most cases of EoCRC are sporadic and associated with environmental exposures. Growing evidence suggests that the gut microbiome may play a critical role in colorectal carcinogenesis, including tumor initiation and progression. This has led to the hypothesis that gut dysbiosis could partially account for the rising incidence of EoCRC. However, the current literature on this topic remains limited. This study aims to systematically map and synthetize the existing evidence on the gut microbiome characterization in EoCRC, with a particular focus on the methodologies employed. By identifying existing knowledge gaps and methodological limitations, this review seeks to inform future research directions and clarify the role of the gut microbiome in EoCRC. Ultimately this may lead to the discovery of modifiable microbial targets, offering novel strategies for the prevention or adjunctive treatment of colorectal cancer and contributing to the reduction in disease burden.

## 1. Introduction

Colorectal cancer (CRC) remains a major global health concern, ranking as the third most commonly diagnosed cancer and the second most lethal worldwide [1]. Over the past two decades, both the USA and Europe have observed a rising incidence of early-onset CRC (EoCRC)—defined as CRC diagnosed before the age of 50 years—a trend that is accelerating and thus constitutes a public health issue [2,3]. In the 1990s, CRC was the fourth cause of cancer-related death in young people, but nowadays it is the leading cause of cancer-related death in young men and the second cause in young women, after breast cancer [4]. By 2030, it is estimated that 10% of CRC and 22% of rectal cancers in the USA will be EoCRC and that CRC will be the first cause of cancer-related-mortality in adults between 20 and 49 years of age [5]. The etiology behind this increasing incidence remains unclear. These patients are typically not included in conventional screening programs. EoCRC patients have a later diagnosis, a more advanced staging, and more aggressive pathology features at diagnosis than late-onset CRC (LoCRC) patients [6,7,8]. This highlights a significant gap in current clinical practice, underscoring the urgent need for tailored screening, diagnostic, and treatment protocols specific to EoCRC.

The vast majority of EoCRC are sporadic and related to environmental exposures [9]. Several risk factors associated with the development of CRC, namely sedentary lifestyles, obesity, smoking, alcohol consumption, and a diet rich in red and processed meat and low in fruits, vegetables, and fiber are also involved in EoCRC risk [10].

The gut microbiome is a complex community of approximately one hundred trillion microorganisms that mainly reside in our colon and play a vital role in human health. Gut microbiota dysbiosis is an imbalance in microbiota composition and function in comparison with healthy individuals. Environmental exposures during a person’s lifetime, referred to as the exposome, may influence the gut microbiome and lead to dysbiosis, which has been implicated in several diseases [11]. Dysbiosis seems to contribute significantly to CRC tumorigenesis by promoting chronic inflammation and immune dysregulation [12,13].

Alpha diversity is a measure of the richness (i.e., number of distinct taxa) or evenness (i.e., relative abundance of taxa) within a microbiota population, while beta diversity is the variability in the identification of taxa observed between microbial populations across different samples. A 2022 systematic review analyzed the studies that compared gut microbiota diversity between CCR and healthy controls and found divergent results regarding α diversity, while β diversity was different in half the studies [14].

Studies comparing the gut microbiota of CRC patients and healthy individuals have consistently identified an increase in pro-inflammatory bacteria and a decrease in beneficial species. *Fusobacterium nucleatum*, some *Escherichia coli* strains, *Bacteroides fragilis*, *Prevotella*, *Enterococcus faecalis*, and *Streptococcus bovis* are increased in CRC patients, whereas *Firmicutes*, *Bifidobacterium*, *Bacteroides vulgaris*, *Lactobacillus*, and *Clostridium* are consistently decreased [12,15,16,17,18,19,20]. Microbiota can also modulate aging-related changes. These findings gave rise to the hypothesis that microbial composition could partially account for the increasing incidence of EoCRC.

To explore this hypothesis, a preliminary search was conducted to identify existing or ongoing systematic or scoping reviews addressing microbiome characterization in EoCRC. No such reviews were identified. Given the emerging nature of the topic and the anticipated heterogeneity in study designs, microbiome analysis methods, and reporting practices, a scoping review was deemed the most appropriate methodological approach to comprehensively map the breadth of available evidence.

## 2. Objectives

This systematic scoping review seeks to answer the following research question: What evidence exists regarding the characterization of the gut microbiome in EoCRC patients? Specifically, it aims to systematically map and synthesize the existing evidence on the gut microbiome characterization in EoCRC, compared to LoCRC and healthy individuals, in both clinical and research settings. Additionally, it seeks to examine the methodology used and identify knowledge gaps in the literature to inform and guide future research in this field. A deeper understanding of the gut microbiota in EoCRC may uncover modifiable microbial targets, potentially offering novel adjunctive strategies for CRC prevention or treatment and ultimately contributing to reducing disease burden.

## 3. Methods

The proposed scoping review is reported in accordance with the Joanna Briggs Institute (JBI) Evidence Synthesis Manual 2024 (Adelaide, Australia) and the Preferred Reporting Items for Systematic Reviews and Meta-Analyses for Scoping Review [21,22]. The protocol has not been registered.

### 3.1. Eligibility Criteria

#### 3.1.1. Participants

We considered studies that included EoCRC patients, i.e., individuals diagnosed with CRC before the age of 50 years. Studies that do not segregate EoCRC from LoCRC or that do not disclose patient’s age at diagnosis were not considered.

#### 3.1.2. Concept

We focused on studies that characterize the gut microbiota, with conclusions regarding its taxonomic composition. Particular attention was given to the analytical techniques and the sampling methods, as these factors critically influence the interpretation and comparability of findings.

#### 3.1.3. Context

We considered clinical and research settings, where microbiome analysis was performed, independently of geographic location, racial, or gender details.

#### 3.1.4. Types of Sources

This scoping review considered both experimental and observational studies, including case series, case reports, prospective and retrospective cohort studies, case–control studies, and cross-sectional studies. Systematic or literature reviews, meta-analyses, guidelines, conference abstracts, animal studies, and in vitro experiments were excluded.

### 3.2. Search Strategy

A three-step search strategy was adopted in this review, which was conducted by two researchers (RGS and CSG) between January and March 2025. First, an initial limited search was undertaken in PubMed to identify the keywords and MESH terms contained in the titles and abstracts of relevant articles on the topic. In a second step, a full search in PubMed, Web of Science, and Scopus was undertaken, using a combination of the following keywords “young” OR “early” AND “microbiome” OR “microbiota” OR “dysbiosis” AND “colorectal cancer” OR “colorectal neoplasm” OR “colorectal carcinoma” OR “gastrointestinal cancer”. We considered only studies published in English. Thirdly, the reference lists of the potentially eligible papers was searched for additional sources. In Appendix A, a full description of the PubMed search is provided. The full search strategy is available from the authors upon request.

### 3.3. Source of Evidence Screening and Selection

The studies resulting from each search were independently screened by two researchers (RGS and CSG), based on title and abstract. Duplicate entries were removed, and the relevant articles were retrieved for full-text reading. The same two researchers independently assessed the full texts of these studies. Any discrepancies in selection were discussed and resolved by consensus.

### 3.4. Data Extraction

The JBI data extraction template was used and adapted to extract relevant information [21]. One researcher (RGS) extracted the following items: author, year of publication, population, aims, sample, source of microbiota sample, sequencing methodology, primary and secondary outcomes measures, main results (diversity, taxonomic characterization, and other results) and conclusions. Data were recorded in an Excel spreadsheet.

### 3.5. Synthesis of Results

For each included study, data were qualitatively synthesized and presented in a tabular form. Results were separated by gut microbiota diversity and composition.

## 4. Results

The PRISMA flowchart for study identification and selection is summarized in Figure 1. The search resulted in the identification of 1375 articles. After duplicate removal (*n* = 681), 694 articles were screened by title/abstract. Of these, 679 were excluded and 15 were considered for retrieval. The reading of the reference list of these potentially eligible articles did not result in any other inclusion. The 15 papers were full-text screened by two reviewers (RGS and CSG), who agreed to exclude eight papers: four included patients with colorectal adenomas, three included CRC patients with early-stage disease, and one was a protocol study. Seven articles met eligible criteria and were included.

### 4.1. Study and Sample Characteristics

Characteristics of included studies are displayed in Table 1 [23,24,25,26,27,28,29].

Of the seven studies included, five comprised a Chinese population and two an American one, and all were published between 2021 and 2024, mirroring the growing global interest and the emerging nature of this topic.

### 4.2. Used Methodology

All seven are case–control studies. In total, 743 EoCRC, with a median sample size of 106 (20–185), were compared with 2261 LoCRC and/or 473 young healthy controls. All studies compared EoCRC with LoCRC; one study did it exclusively. The population age cutoff for LoCRC varied between ≥50 years, ≥55 years (Xiong et al. [27]), >60 years (Barot et al. [24]), and >65 years (Adnan et al. [26]). Adnan et al. excluded these middle-aged adults in order to better distinguish between the two populations, given that they could have overlapping features [26].

Regarding the sequencing methodology: three studies used 16S ribosome RNA (rRNA) gene sequencing [24,27,28], three shotgun sequencing [25,26,29] and one study used both methods [23].

Four studies analyzed fecal samples [23,25,27,29], two studies tumor samples [24,28], and in one study both types of samples were considered [26]. All the tumor samples were obtained from surgery specimens. Of the 3 studies that used tumor samples, Barot et al. compared microbiota from tumor with microbiota from adjacent non-malignant biopsies from EoCRC and LoCRC patients [24].

### 4.3. Gut Microbiota Diversity

All studies examined gut microbiota diversity, but the results are divergent. The studies comparing gut microbiota diversity between EoCRC and young healthy controls are presented in Table 2, while those comparing EoCRC and LoCRC are depicted in Table 3.

Three studies reported decreased α diversity in EoCRC patients [23,27,29], whereas one study found an increased α diversity [25] when compared with young healthy controls. The four studies documented that β diversity was different between EoCRC and young healthy controls [23,25,27,29].

Four studies compared microbiota diversity between EoCRC and LoCRC using fecal samples [23,25,27,29]. One study reported that EoCRC patients had higher α diversity [23], other study observed that EoCRC patients had lower α diversity [27], and two studies found no difference in α diversity between the two groups [25,29]. Two studies verified that they had different β diversity [23,27], but one study documented an equal β diversity [25].

Two studies compared gut microbiota diversity between EoCRC and LoCRC using tumor samples [24,28]. One study found that EoCRC patients had higher α diversity [24], and the other that EoCRC patients had lower α diversity [28]. Both studies documented different β diversity between EoCRC and LoCRC patients [28].

### 4.4. Taxonomic Classification

All studies assessed the abundance of specific microbial taxa. Frequently, certain bacterial taxa appeared to be enriched in EoCRC fecal or tumor samples; however, these findings often did not reach statistical significance in regression analysis. Taxa that reached significance differences between EoCRC and young healthy controls are presented in Table 4 and between EoCRC and LoCRC in Table 5.

Four studies comparing gut microbiota profiles of EoCRC patients and young healthy controls using fecal samples reported that EoCRC patients had an enrichment in the following taxa: *Flavonifractor plautii* [23,29], *Bacteroides vulgatus* [29], *Bacteroides cellulositycus* [29], *Parabacteroides sp CT06* [29], *Vibrio_qinghalensis* [29], *Odoribacter splanchnicus* [29], *Fusobacteria* [27], *Akkermansia muciniphila* [26], *Bacteroides fragilis* [26], *Bacteroides cellulolyticus* [26], *Eubacterium siraem* [26], *Erysipelatochlostridium ramosum* [26], *Oscillibbacter sp.CAG.241* [26], and *Enterocloster boltoes* [26].

Regarding gut microbiota profile comparison between EoCRC and LoCRC patients, one study using fecal samples reported an enrichment in *Fusobacteria* in EoCRC patients [27]. Two studies using tumor samples observed a predominance of *Actinomyces* and *Akkermansia and Bacteroides* in EoCRC patients [24,28].

Only three studies correlated the microbiota composition of EoCRC patients with their clinicopathology data [23,24,25]. Barot et al. verified that EoCRC and LoCRC patients had distinct microbial profiles associated with tumor location, sidedness, TNM stage, and obesity [24]. Among EoCRC patients, those with BMI ≥ 25 kg/m^2^ had a relatively higher abundance of *Limosilactobacillis*, *Listeria*, *Akkermansia*, *Enterococcus*, and *Escherichia/Shigella* genera; when compared with patients with normal BMI. In EoCRC patients, right-sided colon tumors had a higher relative abundance of *Limosilactobacillis*, *Baccilus*, *Akkermansia*, *Pseudomonas*, and *Escherichia/Shigella* than left-sided colon tumors and colon tumors a higher relative abundance of *Limosilactobacillis*, *Staphylococcus*, *Listeria*, *Akkermansia*, and *Pseudomonas* than rectal tumors. Stage I–III tumors were enriched in *Bacteroides* genus when compared to stage IV tumors. However, correlation analysis confirmed only that increased abundance of *Fusobacterium* and *Akkermansia* correlated with a greater likelihood of rectal compared to colon tumors in EoCRC and LoCRC patients. Moreover, *Fusobacterium* and *Akkermansia* abundance negatively and positively correlated, respectively, with overall survival in EoCRC patients. Yang et al. reported that *Alistipes* and *Roseburia* were particularly abundant in left-side colon cancer and right-side colon cancer, respectively, in EoCRC patients [23].

Among these seven studies, only Qin et al. explored correlations between microbiota composition and molecular markers, namely mismatch repair (MMR), *BRAF*, and *HER2* status, but they did not controlled for age [25]. They observed that *Fusobacterium nucleatum* and *animalis* were more abundant in mismatch repair-deficient (dMMR) tumors vs. mismatch repair-proficient (pMMR) tumors and in HER2-overexpressed vs. tumors without HER2 overexpression.

### 4.5. Other Outcomes

Of the three studies that used tumor sampling to study gut microbiome, two complemented the taxonomic classification and comparison with network analysis. Xu et al. verified that EoCRC patients exhibited fewer networks of bacteria [28].

Kong et al. performed a non-targeted metabolomics analysis and observed that EoCRC samples were enriched in several specific amino acids, such as glycine, L-aspartate, tryptophan, microbiota derivates of tryptophan (indole-3-acetaldehyde), bile acids, and choline metabolites [29].

Of the seven studies, six studies included functional analysis. Xu et al. used PICRUS (Phylogenetic Investigation of Communities by Reconstruction of Unobserved States) to infer functional pathways, finding that EoCRC were enriched in pathways related to lipid transport and metabolism [28]. Beyond that, Xu et al. used immunohistochemistry and FISH to correlate the presence of *Actinomyces* with cancer-associated fibroblast, signaling pathways, and lymphocytes patterns and found that *Actinomyces* performed well in identifying EoCRC (in comparison with LoCRC) with an area under the curve (AUC) of 0.747. Qin et al. correlated age with abundance of metaCyc pathways, expression of cutC gene as other virulence factors and toxins and found no difference between EoCRC and LoCRC [25]. Four studies used the Kyoto Encyclopedia of Genes and Genomes database, and three studies also employed the Gene Ontology. Adnan et al. calculated pathway enrichment scores and observed a significantly higher percentage of microbial–host pathway correlation in EoCRC when compared with LoCRC, particularly the sulfur metabolism pathway and several DNA repair pathways [26]. Yang et al. observed a dominance of DNA binding and RNA-dependent DNA biosynthetic process in EoCRC patients [23]. Xiong et al. verified that both EoCRC and LoCRC had a reduced expression of genes involved in transcription, defense mechanisms, inorganic ion transport, and metabolism, cell wall/membrane/envelope biogenesis, membrane transport, and porphyrin and chlorophyll metabolism [27]. Kong et al. documented a different distribution of genes between EoCRC and young healthy controls, with upregulation of KO pldB and a cbh gene axis and choline metabolism, establishing a link between microbiota abundance, microbial KO genes, and metabolites [29].

Three authors constructed microbiome-based classification models to distinguish fecal samples of EoCRC patients from young healthy controls [23,25,29]. Yang et al. developed a random forest classifier model with 60 genera markers and achieved high predictive performance with an AUC of 0.88 in external validation [23]. Kong et al. integrated in their random forest model bacteria markers, KO genes, and metabolites profile and obtained an AUC of 0.78 in external validation [29]. Qin et al. used two algorithms: random forest and least absolute shrinkage and selection operator (LASSO) logistic regression [25]. Using a species-based model, the authors obtained better performance than with a pathway-based model and found similar prediction accuracy for CRC status in both EoCRC and LoCRC, suggesting that both populations have a similar microbial signature.

## 5. Discussion

The incidence of EoCRC is increasing steadily and, aside from hereditary factors, other contributing risk factors remain poorly defined. Emerging evidence points to a potential role of the gut microbiome in colorectal carcinogenesis, raising interest in its characterization among EoCRC patients. In the present systematic scoping review, we sought to synthesize the evidence regarding gut microbiome profiles in EoCRC patients. The evidence base remains limited, with only seven studies identified.

Overall, EoCRC patients are generally characterized by lower α diversity and distinct β diversity when compared with healthy young adults. However, studies comparing EoCRC and LoCRC have reported inconsistent findings regarding α diversity, while there is consensus that β diversity differs between these groups. Comparing with healthy young individuals, EoCRC microbiota is enriched in *Flavonifractor plautii* [23,29], *Bacteroides* [26,29], *Fusobacterium* [27], *Akkermansia muciniphila* [26], *Vibrio_qinghalensis* [29], *Odoribacter splanchnicus* [29], and less known Firmicutes [26,29]. In comparison with LoCRC, EoCRC patients seem to have a higher abundance in *Fusobacterium*, *Akkermansia*, *Bacteroides*, and *Actinomyces* [24,27,28].

### 5.1. Gut Microbiota Diversity

According to our findings, individuals with EoCRC tend to exhibit reduced alpha diversity and distinct beta diversity compared to healthy young adults, indicating a less diverse and compositionally different gut microbiota. When comparing EoCRC with LoCRC, the results are less consistent: while there is no clear agreement regarding differences in diversity within individuals, studies generally agree that their overall microbial composition is different. As biological age increases in healthy individuals, gut microbiome undergoes several changes. Overall gut microbial richness tens to decline, with reductions in taxa such as *Prevotella*, *Faecalibacterium*, *Eubacterium rectale*, *Lachnospira*, *Coprococcus*, and *Bifidobacteria*. These are often replaced by *Akkermansia*, *Butyricimonas*, *Butyricicoccus*, *Christensenellaceae*, *Odoribacter*, *Bilophila*, *Eggerthella*, *Enterobacteriaceae*, *Fusobacteria*, and *Streptococcus* spp. [30]. The hypothesis that differences in gut microbiota between EoCRC and LoCRC patients may reflect biological age rather than cancer-specific mechanisms cannot be ruled out. This age-related shift may confound attempts to attribute microbial differences directly to tumor biology. Nonetheless, the consistent observation that EoCRC patients have a distinct microbiota compared to healthy age-matched controls supports the notion that microbial composition may play a role in EoCRC pathogenesis. These findings strengthens the hypothesis that gut microbiome may represent a modifiable risk factor in the EoCRC carcinogenesis and highlights the potential of microbiome-targeted strategies as adjunctive measures for EoCRC prevention and risk reduction.

Interestingly, Qin et al. integrated their Guangzhou cohort with the previously published Fudan cohort from Yang et al., forming the largest cohort of EoCRC patients reported to date and did not replicate Yang et al.’s findings [23,25]. They observed that EoCRC patients had higher α diversity and that most CRC-associated taxa exhibited concordant changes in EoCRC patients and healthy controls [23,25]. These divergent results may be explained by the sample size, which is larger in Qin et al. study, by the absence of control for confounding factors (such as antibiotics use, diet, previous use of probiotics) in both studies, and, at least in part, by methodological differences: while Qin et al. used shotgun metagenomic sequencing and measured α diversity by the Shannon index, Yang et al. employed 16S rRNA sequencing and measured α diversity by the observed species. Metagenomics aims at studying the structure and function of the microbial genome, called metagenome. The first step is the microbial DNA sequencing, which consists of two possible approaches: 16S rRNA and shotgun metagenomics. In the first method, only the 16S rRNA gene is targeted, amplified and sequenced, providing information regarding the taxonomic microbial community structure at the genus level. This technique allows conclusions regarding the diversity, richness and evenness of the microbiota community, but is limited by the potential bias introduced by the choice of primers used to amplify 16S rRNA [31]. Shotgun metagenomic sequencing approach examines the collective genome from all the microbial species in a given sample, provides a high-resolution taxonomic profiling at the species level but also informs on the genic contribution of each member of the community [32,33]. Shotgun sequencing should be the preferred sequencing methodology, but its use is hindered by the fact that it is time-consuming and expensive. Of the four studies that compared α diversity between EoCRC patients and healthy young controls, two studies used 16S rRNA sequencing, including Yang et al. and two shotgun sequencing, namely Qin et al. and Kong et al. [23,25,27,29]. This latter, using the same sequencing methodology, had a different result than the former, whose sample size was larger, which may explain this difference.

We conclude that the consistent result that EoCRC patients, when compared with healthy young adults, have a distinct microbiota gives strength to this observation. However, the results comparing EoCRC and LoCRC are less consistent—the small sample size and the different sequencing methodologies are a limitation to the interpretation of the study’s conclusions.

### 5.2. Taxonomic Classification

According to our findings, EoCRC patients are enriched in *Fusobacteria* [27], *Actinomyces* [28], *Akkermansia*, and *Bacteroides* [24] when compared with LoCRC patients.

*Fusobacterium nucleatum* is a Gram-negative anaerobic bacterium whose colonization of the gut increases with age [34]. It is largely recognized as a key player in the CRC development and progression; it promotes tumor cell proliferation, facilitates metastases, and alters CRC prognosis by inducing chemoresistance [35]. The association of *Fusobacterium* with EoCRC rectal tumors warrants particular attention, given that more than 70% of EoCRC are located in the rectosigmoid region. These tumors are usually pMMR, without chromosomal instability, with fewer somatic *APC*, *KRAS*, and *BRAF* mutations and hypomethylation features [36]. Interestingly, Qin et al. observed that *Fusobacterium nucleatum* was more abundant in dMMR CRC tumors [25]. However, dMMR CRC are generally associated with improved overall survival, and *Fusobacterium* abundance has been linked to worse prognosis, both in CRC in general and in EoCRC in particular [24,37]. The role of Fusobacterium in EoCRC remains unclear and warrants further investigation, particularly given the development of emerging therapeutic strategies that target *Fusobacterium nucleatum* [38].

*Actinomyces odontolyticus*, like *Fusobacterium*, is an anaerobic bacterium existing principally in the oral cavity that is known to have a role in CRC carcinogenesis initiation [39].

*Akkermansia muciniphila* is a Gram-negative anaerobic bacterium of the *Verrucomicrobiacae* family [40]. It produces mucin-degrading enzymes and utilizes mucins as a nitrogen and carbon source in the mucus layer of epithelium, promoting mucus thickness and preserving gut barrier integrity [40]. In healthy adults, it constitutes approximately 3–5% of the gut microbial community, varying according to age: gut colonization starts in early childhood, peeks during adulthood, and declines in older age [41,42]. Evidence is controversial regarding the role of *Akkermansia muciniphila* in CRC, with findings of abundance both increased and decreased in this context [43]. *Akkermansia muciniphila* is considered to promote CRC formation by inducing intestinal epithelial cell signaling dysregulation and triggering inflammation [43]. *Akkermansia muciniphila* abundance is increased in FOLFOX treated individuals and positively correlated with the therapeutic effect. Similarly, it is enriched in dMMR tumors under PD-1 inhibitors, being regarded as a novel strategy for colon cancer therapy [44,45]. Its role in metabolic health is well-established: numerous studies have demonstrated decreased abundance of *Akkermansia muciniphila* in obesity, and its supplementation has shown efficacy in both preventing and treating obesity [34,46]. In this context, the findings reported by Barot et al. are particularly intriguing. The authors observed higher *Akkermansia muciniphila* abundance in EoCRC patients with BMI ≥ 25 kg/m^2^ a result that appears counterintuitive given the established inverse relationship between *Akkermansia muciniphila* and obesity [24]. The confirmation of these results is of great importance, since they could open new horizons for microbiota manipulation as a preventive or therapeutic strategy in EoCRC. Importantly, approximately 50% of patients diagnosed with EoCRC have excess weight or obesity and a growing body of evidence supports an association between EoCRC and metabolic syndrome, with abdominal adiposity identified as an important risk factor [47]. However, Barot et al. remain the only group to date to have examined the relationship between BMI and gut microbiota composition in EoCRC patients [24].

*Bacteroides fragilis* is a colonic symbiote that accounts for 0.5–1% of the fecal microbiota, whose prevalence is higher in the tissue and fecal samples of patients with CRC compared to healthy controls [48].

Comparing EoCRC patients’ and young healthy controls’ gut microbiota profiles, we found that EoCRC patients had an enrichment in *Flavonifractor plautii* [23,29], *Bacteroides vulgatus* [29], *Bacteroides cellulositycus* [29], *Parabacteroides* sp *CT06* [29], *Vibrio_qinghalensis* [29], *Odoribacter splanchnicus* [29], *Fusobacteria* [27], *Akkermansia muciniphila* [26], *Bacteroides fragilis* [26], *Bacteroides cellulolyticus* [26], *Eubacterium siraem* [26], *Erysipelatochlostridium ramosum* [26], *Oscillibbacter* sp. *CAG.241* [26], and *Enterocloster boltoes* [26].

*Flavonifractor plautii* is a flavonoid-degrading bacterium that has been shown to be more abundant in right-sided CRC tumor, as reported by Liang et al. [49]. Its abundance has been negatively associated with dietary flavonoid intake—compounds commonly found in fruits and vegetables and widely recognized for their cancer-preventive properties [50]. Thus, a higher abundance of *Flavonifractor plautii* in individuals with EoCRC may reflect a lower dietary intake of flavonoids. Given the established role of diet in shaping gut microbiota composition, this observation supports the need for comprehensive assessment of dietary patterns in microbiome EoCRC studies. Gut microbiota composition is age-dependent and is highly responsive to the individual lifetime exposome. Yet, most studies reviewed did not account for key environmental and lifestyle risk factors, such as dietary habits, physical activity, alcohol consumption, circadian rhythm disruption, and antibiotic and AINEs usage. The omission of such variables limits our understanding of the microbiota–EoCRC relationship and may contribute to conflicting findings across studies.

*Odoribacter splanchnicus* is a *Bacteroidete* that demonstrated anti-colorectal cancer activity in mouse models [51].

In conclusion, the identification in EoCRC of bacteria known to have a role in CRC carcinogenesis gives strength to these studies. However, the correlation between microbiota composition and clinical, pathological, and molecular features is limited and controversial, namely the possible relation between *Fusobacterium* and rectal EoCRC tumors and lower survival and between *Akkermansia*, overweight patients, rectal EoCRC, and a positive overall survival. While evidence highlights the potential microbial signatures of CRC, the precise roles of these bacterial species in EoCRC pathogenesis remain to be fully elucidated. Future studies integrating microbial and lifestyle data are essential to uncover the mechanistic links between these taxa and EoCRC.

### 5.3. Used Methodology

With this systematic scoping review, we also aimed to understand the methodology used in EoCRC studies in order to better identify knowledge gaps in the literature.

As far as the method of microbiota sampling is concerned, the studies were uniformly distributed, since four studies used fecal samples [23,25,27,29], two studies tumor samples [24,28], and in one study both types of samples [26]. Collecting fecal samples is practical, inexpensive, little labor-intensive, and well-suited for studies evaluating screening tools [52]. However, as a research method, it has important limitations. It is highly influenced by external factors such as antibiotics, probiotics, steroids, or fecal microbiota transplantation treatment and do not accurately represent tumoral microbiome due to horizontal translocation that corresponds to the transfer of bacteria from the proximal to the distal intestine [52]. In research, tumor samples are more relevant for the understanding of the pathological process, since mucosa-associated microbiota, due to its proximity with the epithelium, is found to have an important impact in mucosa homeostasis and tumorigenesis [53]. However, of the three studies that used tumor samples, all the samples came from surgery specimens. The choice of surgical specimens excludes from studies patients with upfront metastatic disease and rectal tumor, who are submitted to chemo and/or radiotherapy, respectively, as a first line of treatment. Using endoscopic biopsies could overcome this limitation, allowing a broader representation of clinical subgroups. However, colonoscopy preparation itself is a relevant confounder. Various authors report that colonoscopy preparation reduce α and β diversity and results in an increase in the abundance of *Proteobacteria*, *Streptococcaceae*, *Veillonella*, and *E. coli* as well as a decrease in the abundance of *Faecalibacterium* and *Lactobacillaceae*. These changes typically disappear two weeks after colonoscopy [54]. However, two studies that used sodium picosulfate as the bowel-cleansing agent found no difference in microbiota before and after colonoscopy [54].

In conclusion, the seven studies have different methodologies, including the method of microbiota sampling, that condition the confounding factors to be considered. None of the seven studies contemplated the use of antibiotics, probiotics, steroids, fecal microbiota transplantation treatment, or the type of colon preparation.

### 5.4. Knowledge Gaps and Future Research

The composition of the adult microbiome varies substantially across geographic regions, largely reflecting differences in industrialization, ethnicity, and dietary patterns. While populations from the USA and China share some similarities in microbiota profiling, they differ from certain European populations [55]. These variations underscore the importance of using population-specific data when investigating disease-associated microbiome signatures, such as those related to CRC. In fact, Adnan et al. observed that geographic location accounted for the greatest proportion of variance in the microbial composition [26].

Across the seven studies, a total of 743 individuals with EoCRC were compared to only 473 young healthy controls. Identifying a distinct microbiome signature associated with EoCRC could have several important implications. Since gut bacteria are closely linked to dietary and lifestyle patterns, such a signature could help pinpoint specific modifiable risk factors. Additionally, it may serve as a novel biomarker for screening, either as a complement or an alternative to current screening methods (if validated). To leverage the microbiome for risk assessment and screening purposes, comparisons between EoCRC and appropriately matched healthy young controls are required. In addition to being geographically exclusive, most studies on EoCRC microbiota are underpowered, with small sample sizes for both EoCRC and control groups. Thus, large-scale, well-designed studies in underrepresented regions—e.g., Europe—are needed to establish reliable microbiome profiles associated with EoCRC.

Further research is warranted to clarify the roles of *Fusobacteria* and *Akkermansia* in EoCRC, particularly exploring their associations with: (1) clinicopathological features, including tumor location, stage, prognosis; (2) lifestyle factors, such as obesity; and (3) tumor molecular characteristics, notably MMR status, since these remain poorly understood and inconsistently reported.

Standardization of methodology across studies is also necessary. If the primary goal is to identify risk factors associated with EoCRC or develop microbiota-based screening tools, we recommend comparing EoCRC exclusively with young healthy controls using fecal samples analyzed through 16S rRNA sequencing. In this context, it is important to control for potential confounding factors such as recent use of antibiotics, probiotics, corticosteroids, or a history of fecal microbiota transplantation. Conversely, if the research focus is on the role of microbiota in EoCRC with a view toward therapeutic applications, comparisons between EoCRC and LoCRC using tumor samples and shotgun sequencing are more appropriate.

## 6. Conclusions

The present systematic scoping review highlights the limited and heterogeneous evidence regarding the gut microbioma characterization in EoCRC, with only seven studies meeting our inclusion criteria. Overall, EoCRC patients have a predominance of lower α diversity and different β diversity when compared to healthy young adults. Discordant results were obtained regarding diversity when comparing to LoCRC patients. In comparison with LoCRC, EoCRC patients have higher abundance of *Fusobacterium*, *Akkermansia*, *Bacteroides*, and *Actinomyces*. In comparison with healthy young adults, EoCRC patients have higher abundance of *Flavonifractor plautii*, *Akkermansia muciniphila*, *Bacteroide*, and *Fusobacteria*. Important hypotheses have been drawn, namely the possible relation between *Fusobacterium* and rectal EoCRC tumors and lower survival and between *Akkermansia*, overweight patients, rectal EoCRC, and a positive overall survival. However, most studies to date have small sample sizes, heterogeneous sequencing and analysis methods, and a lack of integrated clinical, molecular, and lifestyle data that could clarify the drivers of microbial shifts. Despite these limitations, our findings hopefully highlight the need for further research in this area and provide a roadmap for tailoring future methodologies fot the unclarified key topics in this field.

Through this review, the authors aim to contribute to the growing body of literature and to stimulate further research in this field. A deeper understanding of the gut microbiota’s role in EoCRC may contribute to more effective prevention strategies, the development of microbiota-based screening programs, and the use of microbiota as a novel therapeutic approach. Ultimately, these efforts aim to reduce EoCRC incidence and mortality.

## Figures and Tables

**Figure 1 cancers-17-01863-f001:**
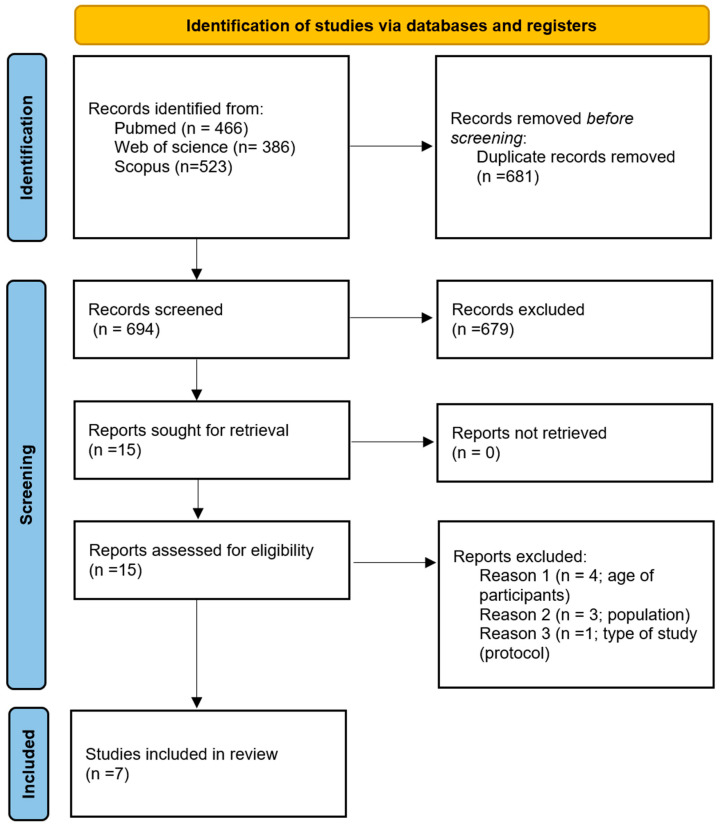
PRISMA flowchart.

**Table 1 cancers-17-01863-t001:** Characteristics of included studies.

	Population	Aim	Sample	Source of Microbiota Sample	Sequencing Methodology	Outcomes
Yang et al. 2021 [23]	China	To evaluate the diagnostic value of gut microbiota for EoCRC patients	185 EoCRC, 379 LoCRC, 217 young controls and 257 old controls.	Stools	16S rRNA gene (all samples) and shotgun sequencing (200 samples, 50 from each group)	Diversity, taxonomic profiling, functional analysis, model construction (random forest).
Kong et al. 2022 [29]	China	To characterize the interactions between gut microbiome, metabolites and microbial enzymes in EoCRC patients and evaluate their potential as non-invasive biomarkers for EoCRC.	Discovery cohort: 114 EoCRC, 130 LoCRC, 97 older controls and 100 young controls. Independent cohort: 24EoCRC, 22 LoCRC and 24 young controls.	Stools	Shotgun sequencing	Diversity, taxonomic profiling, metabolomic composition, functional analysis, model construction (random forest).
Xu et al. 2022 [28]	China	To identify microbial markers for EoCRC diagnosis and explore their potential roles in the tumor immune microenvironment and tumorigenesis.	20 EoCRC, 19 LoCRC. Independent cohort: 78 CRC.	Tumor	16S rRNA sequencing	Diversity, taxonomic profiling, network association, functional analysis, abundance of *Actinomyces* by FISH and correlation with immunohistochemistry.
Xiong et al. 2022 [27]	China	To explore whether there is an alternative gut microbiota profile in patients with EoCRC	24EoCRC, 43 LoCRC and 31 young controls	Stools	16S rRNA gene sequencing	Diversity, taxonomic profiling, functional analysis.
Qin et al. 2024 [25]	China	To address the question whether CRC signatures derived from old patients are valid in young patients	Guangzhou cohort: EoCRC 167 and 293 LoCRC. Fudan cohort: EoCRC 156, 241 LoCRC and 153 young controls.	Stools	Shotgun sequencing	Diversity, taxonomic profiling, functional analysis, model construction (random forest and LASSO logistic regression).
Adnan et al. 2024 [26]	USA	To investigate age-related differences in the gut microbiome of CRC patients and healthy individuals	CuratedMetagenomeData: 82 EoCRC; 1187 LoCRC and 125 young controls. Cancer Genoma Atlas: 15 EoCRC and 70 LoCRC.	CuratedMetagenomeData: Stools. Cancer Genoma Atlas: Tumor	Shotgun sequencing	Diversity, taxonomic profiling, functional analysis.
Barot et al. 2024 [24]	USA	To compare the tumor microbial profile of EoCRC with average-onset CRC and to assess its association with clinical factors	136 EoCRC, 140 LoCRC and 276 adjacent non-malignant sample	Tumor	16S rRNA sequencing	Diversity, taxonomic profiling, network analysis.

**Table 2 cancers-17-01863-t002:** Comparing gut microbiota diversity between EoCRC patients and young healthy controls.

	Measures	α Diversity	Measures	β Diversity
Fecal microbiota				
Yang et al. [23]	Observed species (*p* = 4.12 × 10^−8^)	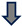	PCoA of weighted UniFrac distances and Permanova (*p* = 0.019)	≠
Kong et al. [29]	Breakaway estimates (*p* = 0.0074)	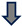	PCoA of Bray–Curtis distance and Permanova (*p* = 0.001)	≠
Xiong et al. [27]	Shannon index (*p* = 0.008)Simpson index (*p* = 0.011)	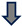	PCoA unweighted UniFrac distances and Permanova (*p* = 0.0001)	≠
Qin et al. [25]	Shannon index (*p* = 0.053 Fudan cohort; *p* = 1.7 × 10^−5^ Guangchou cohort)		PCoA of Bray–Curtis distance and Permanova (*p* = 0.46 Fudan cohort *p* = 0.012 Guangchou cohort)	≠

The red arrow (

) reflects an increase in EoCRC patients; the gray arrow (
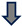
) reflects a decrease in EoCRC patients; and ≠ reflects a difference between groups. PCoA = Principal coordinate analysis.

**Table 3 cancers-17-01863-t003:** Comparing gut microbiota diversity between EoCRC and LoCRC patients.

	Measures	α Diversity	Measures	β Diversity
Fecal microbiota				
Yang et al. [23]	Shannon index (*p* = 8.88 × 10^−5^)		PCoA of Weighted UniFrac distances and Permanova (*p* = 0.001)	≠
Kong et al. [29]	Breakaway estimates (*p* = 0.0788)	=		Not available
Xiong et al. [27]	Shannon index (*p* = 0.007)Simpson index (*p* = 0.013)	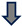	PCoA unweighted UniFrac distances and Permanova (*p* = 0.0001)	≠
Qin et al. [25]	Observed species (*p* = 0.13) Shannon index (*p* = 0.42)	=	PCoA of Bray–Curtis distance and Permanova (*p* = 0.15)	=
Tumor microbiota				
Xu et al. [28]	Chao1 index (*p* = 0.002), ACE index (*p* = 0.003), Shannon index (*p* = 0.063), Simpson index (*p* = 0.673), Pielou-e index (*p* = 0.354)	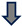	PCoA of unweighted (*p* = 0.051) and Jaccard (*p* = 0.003) UniFrac distances and Permanova	≠
Barot et al. [24]	Shannon index(*p* = 1.5 × 10^−5^)		PCoA of Bray–Curtis distance and Permanova (*p* = 0.013)	≠

The red arrow (

) reflects an increase in EoCRC patients; the gray arrow (
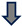
) reflects a decrease in EOCRC patients; = means no difference; and ≠ means a difference between groups. PCoA = Principal coordinate analysis.

**Table 4 cancers-17-01863-t004:** Comparing gut microbiota profile between EoCRC patients and young healthy controls.

Bacteria		Study (Statistical Significance)
Fecal microbiota		
*Flavonifractor plautii*		Yang et al. (LDA score *p* < 0.01), Kong et al. (LDA score *p* < 0.05) [23,29].
*Bacteroides vulgatus*		Kong et al. (LDA score *p* < 0.05) [29]
*Bacteroides cellulositycus*		Kong et al. (LDA score *p* < 0.05) [29]
*Parabacteroides* sp *CT06*		Kong et al. (LDA score *p* < 0.05) [29]
*Vibrio_qinghalensis*		Kong et al. (LDA score *p* < 0.05) [29]
*Odoribacter splanchnicus*		Kong et al. (LDA score *p* < 0.05) [29]
*Fusobacteria*		Xiong et al. (LDA score *p* < 0.001) [27]
*Akkermansia muciniphila*		Adnan et al. (MLR *p* < 0.05) [26]
*Bacteroides fragilis*		Adnan et al. (MLR *p* < 0.05) [26]
*Bacteroides cellulolyticus*		Adnan et al. (MLR *p* < 0.05) [26]
*Eubacterium siraem*		Adnan et al. (MLR *p* < 0.05) [26]
*Erysipelatochlostridium ramosum*		Adnan et al. (MLR p < 0.05) [26]
*Oscillibbacter* sp. *CAG.241*		Adnan et al. [26]
*Enterocloster boltoes*		Adnan et al. (MLR *p* < 0.05) [26]

The red arrow (

) reflects an increase in EoCRC. LDA = linear discriminant analysis. MLR = multivariable linear regression.

**Table 5 cancers-17-01863-t005:** Comparing gut microbiota profile between EoCRC and LoCRC patients.

Bacteria		Study (Statistical Significance)
Fecal microbiota		
*Fusobacteria*		Xiong et al. (LDA score *p* < 0.001) [27]
Tumor microbiota		
*Actinomyces*		Xu et al. (LDA *p* < 0.05) [28]
*Akkermansia*		Barot et al. (*p* = 4.1 × 10^−2^) [24]
*Bacteroides*		Barot et al. (*p* = 4.1 × 10^−2^) [24]

The red arrow (

) reflects an increase in EoCRC patients. LDA = linear discriminant analysis.

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
