# Peer review of "The Knowledge Gap in Gut Microbiome Characterization in Early-Onset Colorectal Cancer Patients: A Systematic Scoping Review"

_cancers, 2025, doi:10.3390/cancers17111863_

Round 1

Reviewer 1 Report

Comments and Suggestions for Authors

This is a good systematic scoping review, through which the authors aimed to deepen the state of the art of the literature and stimulate further research into the role of the gut microbiota in the pathogenesis of early-onset colorectal cancer. Through this systematic scoping review, the authors highlighted the limited and heterogeneous evidence regarding the characterization of the gut microbiome in early-onset colorectal cancer, with only seven studies included. Overall, patients with early-onset colorectal cancer present a predominance of lower alpha diversity and different beta diversity compared to healthy young adults. The authors reiterated their intent to contribute to a deeper understanding of the role of the intestinal microbiota in early-onset colorectal cancer to contribute to effective prevention strategies, namely the development of microbiota-based screening programs and the use of the microbiota as a new therapeutic approach, to reduce the incidence and mortality of early-onset colorectal cancer. Overall, this is an interesting and well-designed work.

Author Response

Thank you very much for your time to review the paper and for your comments.

Reviewer 2 Report

Comments and Suggestions for Authors
  1. "Flavonifractor plautii" should be italicized (Table 4, Page 14).
  2. In the discussion section, Yang et al. reported a significant decrease in α diversity in EoCRC patients, while Qin et al. found an increase in α diversity. The authors only mentioned methodological differences (such as sequencing technologies) but did not explore the potential impacts of sample size, population characteristics (e.g., differences between Chinese and American populations), or confounding factors (such as diet and antibiotic use).
  3. The "↓" symbol in Tables 4 and 5 does not appear in the figure legend. It is recommended to remove the "gray arrow indicating EoCRC reduction" from Table 4.
  4.  Reference 25 has an incorrect format with two entries for 2023; Reference 30 (Kong et al. 2023) is a duplicate of Reference 28. Please verify these references. Reference 35 also has an incorrect format.
  5. On Page 7, Line 169, "Principle coordinate analysis" should be corrected to "Principal coordinate analysis."
  6. The term "BMI ≥25 kg/m²" should be consistent throughout the document, such as on Line 393, Page 13.
  7. It is recommended to standardize the term as "microbiota" or "microbiome," as used on Line 12 and Line 39.

Author Response

Thank you very much for taking the time to review this manuscript. I will answer all your comments:

Comment 1: "Flavonifractor plautii" should be italicized (Table 4, Page 14).

Response 1: Thank you. I have corrected it.

Comment 2: In the discussion section, Yang et al. reported a significant decrease in α diversity in EoCRC patients, while Qin et al. found an increase in α diversity. The authors only mentioned methodological differences (such as sequencing technologies) but did not explore the potential impacts of sample size, population characteristics (e.g., differences between Chinese and American populations), or confounding factors (such as diet and antibiotic use).

Response 2: Both studies are chineses and didn´t account for confounding factors. Qin has a larger sample size. I registered this information in the text.

Comment 3: The "↓" symbol in Tables 4 and 5 does not appear in the figure legend. It is recommended to remove the "gray arrow indicating EoCRC reduction" from Table 4.

Response 3: Thank you. I have corrected it.

Comment 4: Reference 25 has an incorrect format with two entries for 2023; Reference 30 (Kong et al. 2023) is a duplicate of Reference 28. Please verify these references. Reference 35 also has an incorrect format.

Response 4: Thank you. I have corrected it.

Comment 5: On Page 7, Line 169, "Principle coordinate analysis" should be corrected to "Principal coordinate analysis."

Response 5: Thank you. I have corrected it.

Comment 6: The term "BMI ≥25 kg/m²" should be consistent throughout the document, such as on Line 393, Page 13.

Response 6: Thank you. I have corrected it.

Comment 7: It is recommended to standardize the term as "microbiota" or "microbiome," as used on Line 12 and Line 39.

Response 7: I reviewed the paper and used "microbiota" for the bacteria and "microbiome" for all the microbiota environment.

You can find in the reviewed paper highlighted in yellow the changes, with the exception of the references, that were all changed.

Reviewer 3 Report

Comments and Suggestions for Authors

This study starts from the hypothesis that microbiome dysregulation could be responsible for the increasing incidence of CRC. It aims to find evidence regarding the characterization of the gut microbiome in CRC patients and to compare the methodology used to identify the missing elements in the specialized literature.

The article is well organized and demonstrates an interesting approach in the way of selecting the works published so far as well as in the selection of key points that help to define the microbial composition in CRC and in examining the methodology used.

Although the number of works selected for the analysis is very small, the authors managed to define how bacteria such as Fusobacterium nucleatum can influence mismatch repair (dMMR) depending on the expression level of molecules such as HER2.

The study also highlights the role of metabolomic and metagenomic analysis in determining the structure and function of the microbial genome.

The authors emphasize the limitations of their study and show the need for further research to explore the associations between microbiome components with: (1) clinicopathological characteristics, including tumor location, stage, and prognosis, (2) lifestyle factors, such as obesity, and (3) molecular characteristics of the tumor, especially MMR status, as these remain poorly understood.

However, I suggest to the authors a reorganization of chapter 5 Discussion to clearly highlight the strengths and weaknesses (5.1; 5.2; 5.3) that can be extracted from the analysis performed and which are consistent with the study's conclusions.

Overall, the article could be a substantial contribution to the journal. Therefore, I recommend the manuscript for publication after the authors have considered major changes and updates.

Comments on the Quality of English Language

There are paragraphs in this manuscript where the English version could be improved to express the observations more clearly and support the purpose and applicability of this research.

Author Response

Thank you very much for taking the time to review this manuscript and for your comments. 

Regarding your suggestion: "However, I suggest to the authors a reorganization of chapter 5 Discussion to clearly highlight the strengths and weaknesses (5.1; 5.2; 5.3) that can be extracted from the analysis performed and which are consistent with the study's conclusions."

Response: I included in the end of each paragraph (5.1; 5.2; 5.3) of the discussion a conclusion with the strengths and weaknesses, that are further summarized in the conclusion. I do think this makes the text clearer. 

You can find the changes highlighted in yellow in the text.
